# Antimicrobial Activity of Methylene Blue Associated with Photodynamic Therapy: In Vitro Study in Multi-Species Oral Biofilm

**DOI:** 10.3390/pathogens13040342

**Published:** 2024-04-21

**Authors:** Bruno Bueno-Silva, Javier Parma-Garcia, Lucio Frigo, Lina J. Suárez, Tatiane Tiemi Macedo, Fábio Hideaki Uyeda, Marcelo Augusto Ruiz da Cunha Melo, Roberto Sacco, Carlos Fernando Mourão, Magda Feres, Jamil Awad Shibli, Luciene Cristina Figueiredo

**Affiliations:** 1Department of Periodontology, Dental Research Division, Guarulhos University, Guarulhos 07023-070, Brazil; brunobue@unicamp.br (B.B.-S.); javierparmagarcia@gmail.com (J.P.-G.); luciofrigo@uol.com.br (L.F.); lijsuarezlo@unal.edu.co (L.J.S.); magda.feres@hsdm.harvard.edu (M.F.); jshibli@ung.br (J.A.S.); lucienedefigueiredo@gmail.com (L.C.F.); 2Faculdade de Odontologia de Piracicaba, Universidade Estadual de Campinas, Piracicaba 13414-903, Brazil; 3Faculdade de Odontologia da Associação Paulista de Cirurgiões Dentistas (FAOA), São Paulo 02011-000, Brazil; 4Departamento de Ciencias Básicas y Medicina Oral, Facultad de Odontología, Universidad Nacional de Colombia, Cra 45 # 26-85, Bogotá 11001, Colombia; 5Department of Oral Surgery, Faculty of Dentistry, Oral & Craniofacial Sciences, King’s College London, London SE1 9SP, UK; 6Department of Periodontology, Dental Research Division, Tuft University School of Dental Medicine, Boston, MA 02111, USA; carlos.mourao@tufts.edu; 7Department of Oral Medicine, Infection, and Immunity, Division of Periodontology, Harvard School of Dental Medicine, Boston, MA 02115, USA

**Keywords:** photodynamic therapy, methylene blue, biofilms, periodontal diseases, peri-implantitis, laser

## Abstract

The control of infectious diseases caused by biofilms is a continuing challenge for researchers due to the complexity of their microbial structures and therapeutic implications. Photodynamic therapy as an adjunctive anti-infective treatment has been described as a possible valid approach but has not been tested in polymicrobial biofilm models. This study evaluated the effect of photodynamic therapy in vitro with methylene blue (MB) 0.01% and red LEDs (λ = 660 nm, power density ≈ 330 mW/cm^2^, 2 mm distance from culture) on the metabolic activity and composition of a multispecies subgingival biofilm. Test Groups LED and MB + LED showed a more significant reduction in metabolic activity than the non-LED application group (~50 and 55%, respectively). Groups LED and MB equally affected (more than 80%) the total bacterial count in biofilms. No differences were noted in the bacterial biofilm composition between the groups. In vitro LED alone or the MB + LED combination reduced the metabolic activity of bacteria in polymicrobial biofilms and the total subgingival biofilm count.

## 1. Introduction

Microbial biofilms are one of the leading causes of infection in the human body. Furthermore, they are related to the most significant number of deaths from infectious diseases at the hospital level [1]. In the oral cavity, dental biofilms have been identified as the cause of microbial-induced diseases of a different nature, in which the presence of dysbiosis of bacterial communities contained in the biofilm not only alters the balance with the host but the metabolism of nutrients as well. This gives rise to multiple diseases, among them caries, and diseases in the tissues of protection and support at periodontal and peri-implant levels [2,3,4].

Biofilms are characterized by having complex interactions with the surface where they are formed and, therefore, with the microorganisms on it. Biofilms are microbial conglomerates surrounded by an extracellular matrix of polymeric substances such as exopolysaccharides, proteins, and nucleic acids, providing them with protection from the action of antimicrobials of any nature. Moreover, this matrix allows inter-bacterial communication and the mobilization of chemicals and nutrients through the generation of diffusion gradients [5,6]. This suggests that the treatment of diseases induced by biofilms has different connotations than those caused by planktonic bacteria and that the treatment of biofilm-induced diseases must include the initial disruption of these communities. 

Bearing this in mind, multiple therapies have been suggested in addition to mechanical disruption for the treatment of oral diseases caused by biofilms, with the aim of gaining direct control of the infectious agent [1]. In the literature, different treatments have been proposed for decontaminating teeth and implant surfaces as adjunctive therapies in treating periodontal and peri-implant diseases of bacterial origin [7,8]. These adjunctive therapies include the use of antibiotics and antiseptics [9,10], lasers [11,12], biological approaches such as the use of prebiotics and probiotics [13,14], the use of phages and Lysins [15,16,17], and dietary interventions [18], among others. Furthermore, in the search for new strategies for microbial control in addition to mechanical therapy, photodynamic therapy has been described as a possible approach. It has, however, shown contradictory preliminary results that cannot be underestimated since both clinical and microbiological outcomes seeking to prove its effect could be deviated by the designs of studies [19,20,21,22]. 

Photodynamic therapy (PDT) consists of a light source with a specific wavelength that is used to stimulate a substance capable of absorbing light of a specific wavelength and transforming it into valuable energy. This substance is called photosensitizing dye, which absorbs light, and this stimulation allows the photosensitizing dye molecules to change from a latent singlet state to an excited triplet state. As a result, free oxygen and molecules react together to allow the formation of reactive oxygen species that promote the destruction of bacterial cells [11,23]. 

Although the scientific literature has demonstrated a beneficial clinical effect of PDT as single or adjunctive therapy in cases of biofilm-caused diseases such as periodontitis and peri-implantitis, the direct antimicrobial activity has been only been proven for some oral bacterial species [24,25,26], mono-microbial biofilms [27], or non-characterized in vivo biofilms [28,29]. 

The effect of photodynamic therapy on multispecies biofilms has been poorly studied, although there are indications of its clinical efficacy in the treatment of infections caused by them. Ex vivo studies have proven the effectiveness of approaches with methylene blue and benzalkonium chloride plus light activation with iodine laser, demonstrating significant reductions of biofilms in endotracheal tubes after a single treatment, with complete eradication of microorganisms (the main ones being *Streptococcus* and *Staphylococcus* species as well as *Candida albicans* and *Pseudomonas aeruginosa*) in 65% of the tubes [30]. In oral environments, it has been shown that in biofilms of *C. albicans* and *S. sanguinis*, photodynamic therapy with erythrosine used as a photosensitizer (400 μM for 5 min) and LED light (532 ± 10 nm for 3 min), causes significant reductions, being greater in monomicrobial biofilms of the microorganisms mentioned [31]. 

Moreover, several studies [32,33,34] have chosen and justified the use of LED lights with wavelengths between 450–660 nm, which comprise the spectrum used in the present study. The reasons that support this are the physical characteristics of the LED, including its narrow emission spectrum, which benefits the maximum absorption of the photosensitizers. Furthermore, the use of a LED source guarantees a larger irradiation field due to the lack of collimation and perfect coherence of the LEDs [35,36]. 

As a low-cost therapy without any side effects described, an in vitro study was developed to evaluate the effect of photodynamic therapy with methylene blue and red LEDs on the metabolic activity and composition of a multispecies subgingival biofilm.

## 2. Materials and Methods

### 2.1. Subgingival Biofilm Model

The following species were used for subgingival biofilm formation: *Actinomyces naeslundii* ATCC 12104, *Actinomyces oris* ATCC 43146, *Actinomyces gerencseriae* ATCC 23840, *Actinomyces israelii* ATCC 12102, *Veillonella parvula* ATCC 10790, *Actinomyces odontolyticus* ATCC 17929, *Streptococcus sanguinis* ATCC 10556, *Streptococcus oralis* ATCC 35037, *Streptococcus intermedius* ATCC 27335, *Streptococcus gordonii* ATCC 10558, *Streptococcus mitis* ATCC 49456, *Aggregatibacter actinomycetemcomitans* ATCC 29523, *Capnocytophaga ochracea* ATCC 33596, *Capnocytophaga gingivalis* ATCC 33624, *Eikenella corrodens* ATCC 23834, *Capnocytophaga sputigena* ATCC 33612, *Campylobacter showae* ATCC 51146, *Eubacterium nodatum* ATCC 33099, *Fusobacterium nucleatum vincentii* ATCC 49256, *Parvimonas micra* ATCC 33270, *Fusobacterium nucleatum polymorphum* ATCC 10953, *Fusobacterium periodonticum* ATCC 33693, *Prevotella intermedia* ATCC 25611, *Streptococcus constellatus* ATCC 27823, *Porphyromonas gingivalis* ATCC 33277, *Tannerella forsythia* ATCC 43037, *Streptococcus anginosus* ATCC 33397, *Streptococcus mutans* ATCC 25175, *Selenomonas noxia* ATCC 43541, *Propionibacterium acnes* ATCC 11827, *Gemella morbillorum* ATCC 27824, *Eubacterium saburreum* ATCC 33271, and *Campylobacter gracilis* ATCC 33236. 

*Streptococcus* subsp. and *Fusobacterium* subsp. were grown on tryptone soy agar with 5% sheep blood under anaerobic conditions (85% nitrogen, 10% carbon dioxide, and 5% hydrogen), while *Eubacterium nodatum* was cultured on 5% sheep blood fastidious anaerobic agar. *P. gingivalis* was grown on yeast tryptone soy agar enriched with 1% hemin, 5% menadione, and 5% sheep blood. In comparison, *T. forsythia* was grown on yeast tryptone soy agar enriched with 1% hemin, 5% menadione, 5% sheep’s blood, and 1% N-acetylmuramic acid. After 24 h of growth, all species were transferred to 15 mL conical tubes (Falcon^®^ Tubes) with BHI Culture Medium (Brain Heart Infusion, Becton Dickinson, Sparks, MD, USA) supplemented with 1% hemin. 

After 24 h of growth in BHI broth with 1% hemin, the optical density (OD) at 600 nm was adjusted to 0.1, corresponding to approximately 10^8^ cells/mL of each species. Single-cell suspensions from each species were diluted to 10^7^ cells/mL, adjusted for their respective cell sizes. Aliquots of 100 µL containing 10^6^ cells of each species were mixed to obtain a final biofilm inoculum. A quantity of 11 mL of BHI broth with 1% hemin and 5% sheep blood was added to obtain a final biofilm inoculum in a volume of 15 mL. 

The multispecies biofilm model was developed using the Calgary Biofilm Device (CBD). In a 96-well plate (Nunc system; Thermo Scientific, Roskilde, Denmark), 150 μL of inoculum per well was added. The inoculum contained 10^4^ cells of each species, and a plate cover contained the polystyrene pins. The plates were then incubated at 37 °C under anaerobic conditions. After 72 h of incubation, the culture medium was changed, and the samples were transferred to new 96-well plates with fresh broth (BHI broth with 1% hemin and 5% sheep blood) for 4 days. 

### 2.2. Biofilm Treatments 

Biofilm treatment with vehicle (Group CONTROL), methylene blue 0.01% (Group MB), LED (λ = 660 nm, power density ≈ 330 mW/cm^2^, 2 mm distance from culture) (Group LED) and methylene blue associated with LED (Group MB + LED) was performed for 5 min on the last day of biofilm formation (sixth day of growth). The use of LED in the two last groups was performed under the same conditions. After 7 days of biofilm formation, the collection was made, and the microbiological analysis was performed. The experiments were performed in triplicate for each group [37]. 

### 2.3. Biofilm Metabolic Activity 

The percentage reduction in biofilm metabolic activity was determined using 2,3,5-triphenyl tetrazolium chloride (TTC) (catalog no. 17779; Fluka Analytics) and spectrophotometry to differentiate between metabolically active and inactive cells. Pins collected from the biofilm growth phase were washed once with PBS and transferred to plates with 200 μL per well of fresh BHI medium containing 1% hemin with 10% of a 1% TTC solution. Plates were then incubated at 37 °C, under anaerobic conditions, for 6–8 h. TTC conversion was read at 485 nm using a fluorescence spectrophotometer [37]. 

### 2.4. DNA-DNA Hybridization (Checkerboard DNA-DNA) 

Genomic probes for the 33 bacterial species associated with periodontal health and disease were prepared as follows in the literature [38]. All strains were purchased freeze-dried from the ATCC (American Type Culture Collection, Rockville, MD, USA) or the Forsyth Institute (Boston, MA, USA). 

Three 7-day biofilm-coated pins of each group were transferred to Eppendorf tubes containing 100 μL of TE buffer (10 mM Tris-HCl, 1 mM EDTA [pH 7, 6]), and then 100 μL of 0.5 M NaOH were added. After boiling the tubes containing the pins and the final solution for 10 min, 0.8 mL of 5 M ammonium was added to neutralize the solution. The samples were analyzed individually for the presence and quantity of the 33 bacterial species using the DNA-DNA hybridization technique. 

Upon lysis, the DNA was plated onto a nylon membrane using a Minislot device (Immunetics, Cambridge, MA, USA). Once the DNA was attached to the membrane, it was placed in a Miniblotter 45 (Immunetics). The DNA probes Digoxigenin labeled were hybridized to individual lanes of Miniblotter 45. After washing the membranes, DNA probes were detected using a specific antibody to digoxigenin conjugated to phosphatase alkaline. AttoPhos substrate (Amersham Life Sciences, Arlington Heights, IL, USA) was used to detect the signals, and Typhoon Trio Plus (Molecular Dynamics, Sunnyvale, CA, USA) was used to obtain the results, which were then converted to absolute counts compared with the patterns on the same membrane. Standards with 10^5^ and 10^6^ cells from each species were placed in two lanes for each race. A zero record was interpreted as a failure to detect a signal. Comparisons of values after treatment and negative controls were made. To calculate the mean counts of individual bacterial species, the method detection limit was established at 1 × 10^4^; any value under the limit was considered zero. 

### 2.5. Statistical Analysis

The experiments were made in triplicate and then evaluated for each treatment group. The microbiological analysis was expressed as counts (levels) of the 33 bacterial species assessed. Significant differences between the two groups were evaluated using the Kruskal–Wallis test, followed by Dunn’s post hoc test for metabolic activity data and checkerboard data (*p* ≤ 0.05). The statistical significance was set at 5%. Statistical analyses were performed with SPSS 11.0 (IBM Corporation, Armonk, NY, USA). 

## 3. Results

Data analysis (Kruskal–Wallis and Dunn tests) indicated that the Test Groups LED and MB + LED showed a more significant reduction in metabolic activity compared with Groups MB and CONTROL (*p* ≤ 0.05, Figure 1). 

Relative to the total bacterial count obtained by Checkerboard DNA-DNA hybridization, all three test groups (MB, LED, and MB + LED) demonstrated the lowest levels of bacteria when compared with the CONTROL group (*p* ≤ 0.05), without statistical difference between treatments with LED, MB, and MB +LED (*p* ≥ 0.05, Figure 2). 

As regards the composition of the subgingival biofilm, comparisons between each test group and the control group relative to composition showed only a few bacterial species with a statistical difference, these being *V. parvulla* for the MB group, *A. actinomycetemcomitans* for the MB + LED group, and *E. nodatum* for Group LED (Figure 3).

## 4. Discussion

The present in vitro study evaluated the effect of photodynamic therapy with methylene blue and red LEDs on the metabolic activity and composition of an oral multispecies subgingival mature biofilm associated with periodontal and peri-implant diseases. The proposed treatment statistically reduced the biofilm’s metabolic activity and the microorganism levels without altering the microbial composition. 

This is highly relevant since studies that evaluate the therapeutic effects of PDT with the application of light and different photosensitizers, together or separately, in polymicrobial biofilms are rare, which undoubtedly impacts the results of PDT. 

The most prevalent diseases with infectious origin in the oral cavity—caries, periodontitis, and peri-implantitis—are associated with the development of dysbiotic polymicrobial biofilms [39,40,41]. Therefore, the therapeutic approach to a disease caused by biofilms requires taking into account the characteristics of these structures, which make it different from the approach to infection by planktonic bacteria. Among these structural characteristics, we need to consider the fact that the bacteria are embedded in a matrix of polysaccharides, many of them insoluble, which gives them structural integrity and makes them very stable and “resistant” to antimicrobials, UV light, extreme conditions of pH, temperature, salinity, pressure, and low amounts of nutrients [42]. 

There are multiple reports of the effect of PDT on oral microorganisms in a planktonic state [43] or on mono-microbial biofilms [44,45,46] on different surfaces (including titanium alloys). In summary, MB in concentrations as low as 50 µg/mL inhibited more than 90% of viable *S. mutans*, *P. gingivalis*, and *A. actinomicetemcomitans* after 60 s of application [44]. These reports do not reflect what happens in the oral cavity at the level of tooth and implant surfaces with established disease. The fact that bacteria structured in a biofilm are practically shielded from attack by external agents explains the vast majority of the results of the present experiment. It reflects the reality of what happens in the oral cavity. However, studies with planktonic bacteria may be helpful to reveal mechanisms of action. In this sense, the literature reports that as the photosensitizer pH increases, greater *P. gingivalis* inhibition may be found [44]. Further, MB may bind to the *P. gingivalis* cell surface, and then, after absorption of the light (600–700 nm wavelength) with a peak, the photosensitizer produces reactive oxygen species and injures several bacterial proteins, lipids, and carbohydrates, leading to bacterial cell death [47]. Thus, future studies should confirm whether these mechanisms still occur within subgingival biofilms. 

In the present study, MB was used as a photosensitizer. Moreover, in medicine, MB, or methylthioninium chloride, a cationic thiazine dye, has been tested as a drug in the treatment of multiple infectious diseases. MB is a drug, approved by the FDA (Food and Drug Administration) and the EMA (European Medicines Agency), which has a perfect safety profile [48]. Furthermore, it has been tested in infections of a different nature, from malaria [49] to COVID-19 [50], with promising results. In fact, in viruses, their broad-spectrum virucidal activity was determined in the presence of UV light. In addition to its efficacy to inactivate viruses in blood products before being transfused, MB in very low concentrations (micromolar) has displayed virucidal preventive or therapeutic activity against influenza virus H1N1 and SARS-CoV-2 in the absence of UV activation [48]. In bacterial infections, efficacy has been demonstrated on isolates of *M. tuberculosis* with a bacteriostatic effect [51], *Staphylococcus aureus*, *Staphylococcus epidermidis*, and *Klebsiella pneumoniae* [52], and it is widely used in toxicology for the treatment of methemoglobinemia [53].

In searching for alternatives that can be applied clinically, it is essential to emphasize that the light source and the photosensitizer used are agents that do not have cellular side effects or alter the treated surfaces. It has been reported that LED light with a power density similar to the appliance used in the present study (λ 635 nm) has detoxifying effects without causing thermal damage or morphological changes in the irradiated titanium oxide surface. LED light probably maintains the osteoconductive properties of dental implants. Another positive point is that PDT, LED light, and photosensitizers are minimally invasive procedures that can be safely repeated until the desired effects are achieved [45]. 

In the present investigation, therapies were only tested as disruptive agents, and their efficacy was not considered total. However, studies by other groups in different models have found that PDT is associated with the ability to inhibit adherence to surfaces, thus interrupting the initial stages of biofilm formation. The biofilm removal around teeth and implants must be mechanically driven using currents, ultrasound, and air jet prophylaxis, and then associated with the PDT. Moreover, PDT altered biofilm formation [54] and interfered with additional essential functions in the survival of biofilms, such as bacterial quorum sensing [55]. Although this has not been tested in oral biofilms, the background led us to think that PDT should not be used to disrupt the mature biofilm as a primary treatment of periodontitis and peri-implantitis (which are dependent on the presence of dysbiotic biofilms). Instead, PDT could be more useful in maintenance programs after dental biofilm disruption to prevent bacterial re-colonization and should be associated with professional plaque control and oral hygiene (patient compliance).

Under the conditions of the present experiment with a single application of red LED light (λ = 660 nm, power density ≈ 330 mW/cm^2^, 2 mm away from the culture, for 5 min) on the last day of biofilm formation (sixth day of growth), that is, on fully structured biofilms, despite a decrease in metabolic activity (LED and MB + LED) and a decrease in the total numbers of bacteria with the test therapies (MB, LED, and MB + LED) and mild changes in biofilm composition, these therapies are not successful in destroying established attached polymicrobial biofilms, and it supports the concept that multi-targeted therapeutic approaches must treat biofilm driven disease and that all adjuvant therapies must be preceded by the mechanical disorganization of these bacterial structures [6]. 

In different types of infections, it has been shown that photodynamic therapy is strain- and photosensitizer-dependent [56] and that it has a cumulative effect with the mixture of other antimicrobials [57]. A factor that must be taken into account and that can increase the antimicrobial activity of the photosensitizer is the incubation/pre-irradiation time, since this could increase the joint effect of the MB + LED [58]. In addition, the light source must be as close as possible to the photosensitizer to be able to activate the molecules of the substance. 

As most of the evidence has concluded that PDT could be developed in the future as a new therapeutic alternative for diseases caused by biofilms, it is essential not to underestimate the effect of PDT. Thus, it will be necessary to continue carrying out controlled studies with different photosensitizers associated with other antimicrobials, with varying sources of light, and with repeated applications.

## 5. Conclusions

The association of MB + LED or LED alone reduced the metabolic activity and the total counts of a multispecies subgingival biofilm in an in vitro study. However, further studies should evaluate the clinical efficacy of this procedure associated with mechanical disruption of the dental biofilm.

## Figures and Tables

**Figure 1 pathogens-13-00342-f001:**
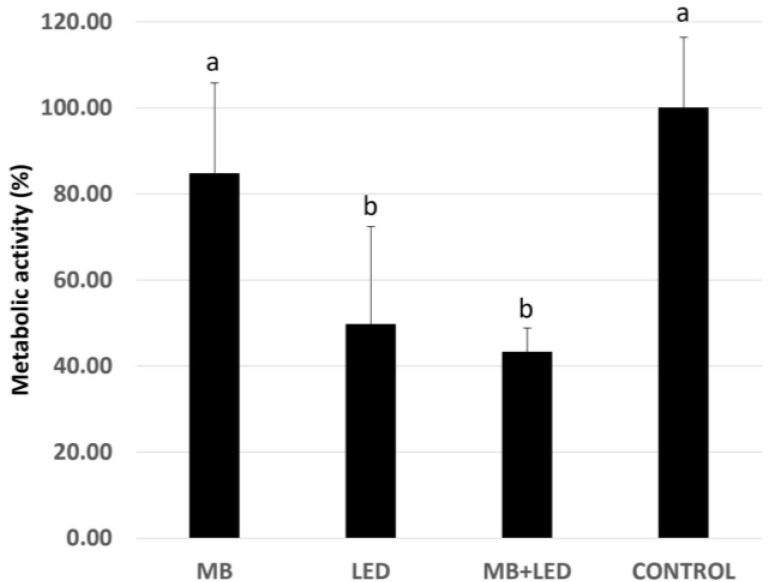
Metabolic activity of biofilms treated with methylene blue 0.01% (MB), with LED, methylene blue associated with LED (MB + LED), and treated with culture media (CONTROL). The results were normalized to those of the control group. Different letters represent statistically significant differences among groups MB and control (letter “a”) and groups LED and MB + LED (letter “b”) by Kruskal–Wallis and Dunn’s posthoc tests (*p* ≤ 0.05).

**Figure 2 pathogens-13-00342-f002:**
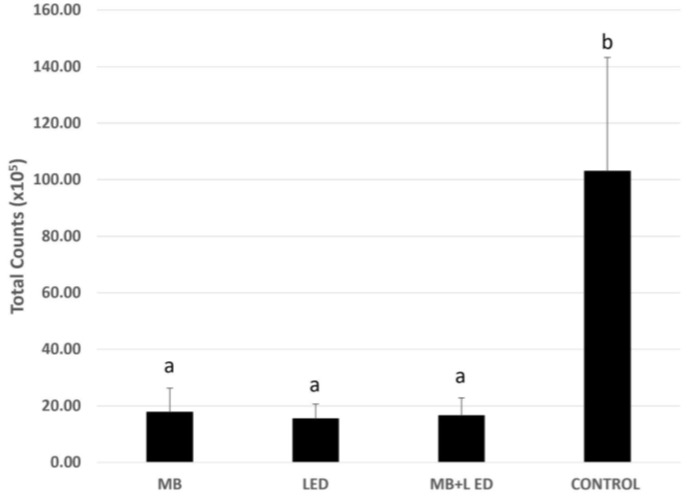
Total counts (×10^5^) of biofilms treated with methylene blue 0.01% (MB), with LED, methylene blue associated with LED (MB + LED), and treated with culture media (CONTROL). Different letters represent statistically significant differences among the control group (letter “b”) and the tested groups (letter “a”) by Kruskal–Wallis and Dunn’s posthoc tests (*p* ≤ 0.05).

**Figure 3 pathogens-13-00342-f003:**
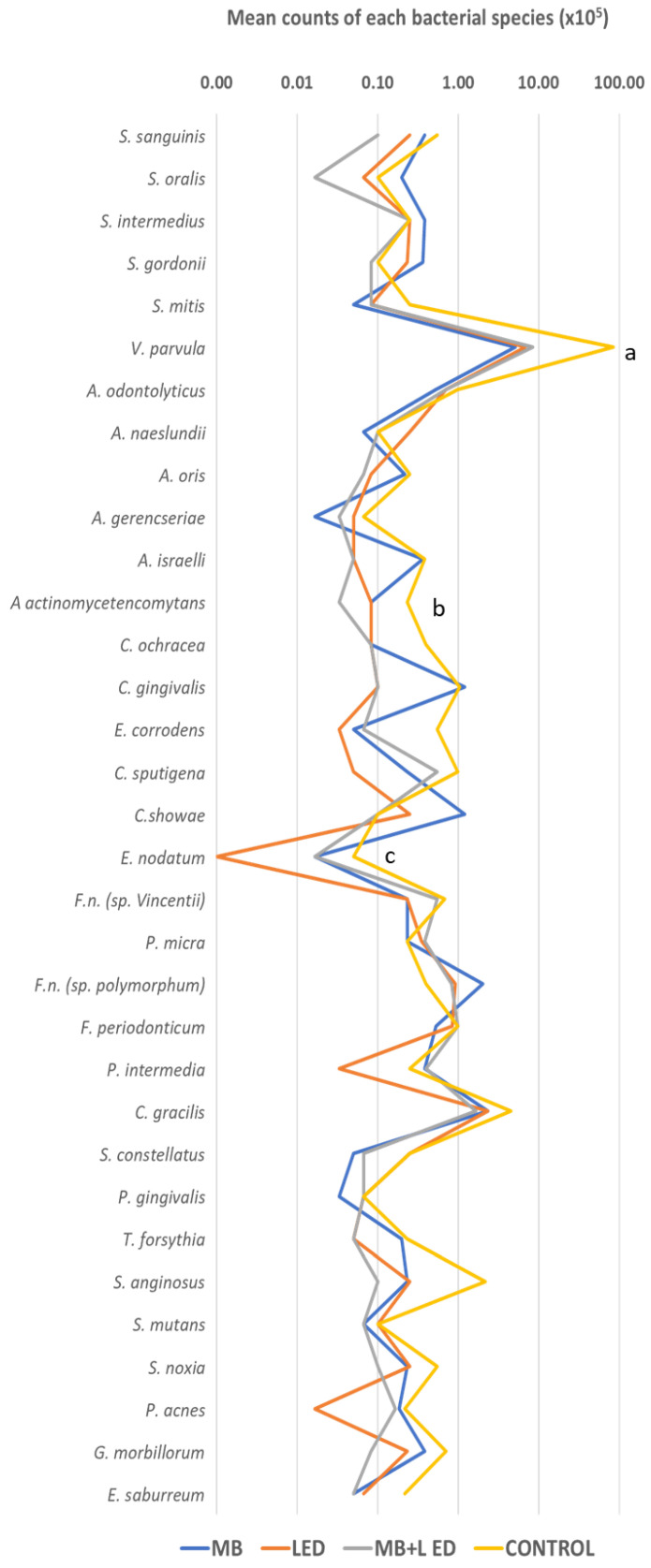
Statistical analysis performed using the Kruskal–Wallis test, followed by Dunn’s post hoc test (*p* ≤ 0.05). The letter “a” means the statistical difference between the biofilm of the blue group and the biofilm of the control group. The letter “b” represents the statistical difference between the biofilm of the blue group + laser for the biofilm of the control group. Finally, the letter “c” represents the statistical difference between the biofilm of the laser group and the control group biofilm.

## Data Availability

Data will be available on request.

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
