# Peer review of "Antimicrobial Activity of Methylene Blue Associated with Photodynamic Therapy: In Vitro Study in Multi-Species Oral Biofilm"

_pathogens, 2024, doi:10.3390/pathogens13040342_

Round 1
Reviewer 1 Report
Comments and Suggestions for Authors
I have reviewed the manuscript entitled: “Antimicrobial Activity Of Methylene Blue Associated With Photodynamic Therapy: In Vitro Study In Multi-species Oral Biofilm”. The authors made a great effort to demonstrate the effectiveness of photodynamic therapy using methylene blue as a photosensitizing dye on biofilms comprised of 33 oral bacteria species, assessing both its impact on metabolic activity and alterations in bacterial composition. However, a significant limitation of the study lies in using a single photosensitizing agent. In addition, the authors did not provide information about prior assays to establish the conditions they used for their single concentration of photosensitizing agent (0.01%) and LED treatment (λ = 660, power density ≈ 330 mW/cm2, 2 mm distance from culture). Also, some sections must be improved (these suggestions can be found in additional comments and the attached file).
Additional comments:
Abstract
Please include the quantitative data such as dosage, percentage of reduction, etc.
Introduction
The introduction provides a clear and concise overview of the significance of microbial biofilms, particularly in the context of oral health, and highlights the challenges they pose in terms of disease development and treatment resistance. The introduction effectively establishes the relevance of biofilms as the major cause of infection and mortality in healthcare settings. Additionally, it elucidates the complexity of biofilm structures, emphasizing their protective mechanisms and implications for disease management. Moreover, the introduction outlines various therapeutic approaches beyond mechanical disruption, reflecting a comprehensive understanding of the need for multifaceted strategies to combat biofilm-associated diseases. However, while the introduction mentions the potential of photodynamic therapy (PDT) as a treatment option, it does not sufficiently contextualize its significance within the broader landscape of biofilm management.
In addition, there is no justification for selecting methylene blue as the photosensitizer agent. As a suggestion, I recommend discussing previous studies that applied similar approaches to combat oral biofilms in more detail, mentioning treatment conditions and the reductions (in biomass, total counts, etc.) achieved.
Materials and methods
Please include a statistical analysis section outlining the experimental design, evaluated factors, response variables, and comparison test used.
Results
The graph in Figure 3 needs a more descriptive title.
Discussion
In lines 225 – 236, the virucidal effect of methylene blue is described, and no apparent relation is established with the results observed in bacterial biofilms. On the other hand, their effect on bacteria is only mentioned without comparison with the presented results and a description of the mode of action. Despite planktonic cells and biofilm former ones presenting different susceptibilities to antimicrobial agents, studies performed with planktonic cultures provide insight into their mode of action. Thus, giving more detail of the mentioned studies in lines 234 – 236 can help explain the observed reduction in the metabolic activity of biofilm-formed cells. This observation also applies to the effect of LED treatment and its combination with methylene blue; thus, it is recommended to include and discuss the outcomes of studies that implemented similar approaches in oral bacteria.
Lines 266 – 274 and 283 – 289 contain the same information described in lines 257-265 and 276-289, respectively.

Author Response
REVIEWER #1
I have reviewed the manuscript entitled: “Antimicrobial Activity Of Methylene Blue Associated With Photodynamic Therapy: In Vitro Study In Multi-species Oral Biofilm”. The authors made a great effort to demonstrate the effectiveness of photodynamic therapy using methylene blue as a photosensitizing dye on biofilms comprised of 33 oral bacteria species, assessing both its impact on metabolic activity and alterations in bacterial composition.
- However, a significant limitation of the study lies in using a single photosensitizing agent. In addition, the authors did not provide information about prior assays to establish the conditions they used for their single concentration of photosensitizing agent (0.01%) and LED treatment (λ = 660, power density ≈ 330 mW/cm2, 2 mm distance from culture).
Author´s Answer: We thank the reviewer for the comment. The reason to use LED lights was included in the following paragraph:
Moreover, several studies [32–34] have chosen and justified the use of LED lights with wavelengths between 450-660nm, which comprise the spectrum used in the present study. The reasons that support this are the physical characteristics of the LED, including its narrow emission spectrum that benefits the maximum absorption of the photosensitizers. Furthermore, the use of a LED source guarantees a larger irradiation field due to the lack of collimation and perfect coherence of the LEDs [35,36].
Also, some sections must be improved (these suggestions can be found in additional comments and the attached file).
Additional comments:
2.Abstract
Please include the quantitative data such as dosage, percentage of reduction, etc.
Author´s Answer: Thanks for the suggestion that improved our manuscript. The requested information was added to abstract.
- Introduction
The introduction provides a clear and concise overview of the significance of microbial biofilms, particularly in the context of oral health, and highlights the challenges they pose in terms of disease development and treatment resistance. The introduction effectively establishes the relevance of biofilms as the major cause of infection and mortality in healthcare settings. Additionally, it elucidates the complexity of biofilm structures, emphasizing their protective mechanisms and implications for disease management. Moreover, the introduction outlines various therapeutic approaches beyond mechanical disruption, reflecting a comprehensive understanding of the need for multifaceted strategies to combat biofilm-associated diseases. However, while the introduction mentions the potential of photodynamic therapy (PDT) as a treatment option, it does not sufficiently contextualize its significance within the broader landscape of biofilm management.
Author´s Answer: We thank the reviewer for his comments. We have added the following information on the effect of PDT on microbial biofilms.
The effect of photodynamic therapy on multispecies biofilms has been poorly studied, although there are indications of its clinical efficacy in the treatment of infections caused by them. Ex vivo studies have proven the effectiveness of approaches with methylene blue and benzalkonium chloride plus light activation with iodine laser, demonstrating significant reductions of biofilms in endotracheal tubes after a single treatment, with complete eradication of microorganisms (the main ones being Streptococcus and Staphylococcus species as well as Candida albicans and Pseudomonas aeruginosa) in 65% of the tubes [30]. On oral enviroments it has been shown that in biofilms of C. albicans and S. sanguinis, photodynamic therapy with erythrosine used as a photosensitizer (400 μM for 5 min) and LED light - 532 ± 10 nm for 3 min, cause significant reductions, being greater in monomicrobial biofilms of the microorganisms mentioned [31].
4.In addition, there is no justification for selecting methylene blue as the photosensitizer agent. As a suggestion, I recommend discussing previous studies that applied similar approaches to combat oral biofilms in more detail, mentioning treatment conditions and the reductions (in biomass, total counts, etc.) achieved.
Author´s Answer: As suggested, we included in the discussion the information requested.
In summary, MB in concentrations as low as 50 µg/mL inhibited more than 90% of viable S. mutans, P. gingivalis and A. actinomicetemcomitans after 60 seconds of application [44].
5.Materials and methods
Please include a statistical analysis section outlining the experimental design, evaluated factors, response variables, and comparison test used.
Author´s Answer: Thanks for the suggestion. The requested information regarding statistical analysis were included as follow:
2.4 Statistical analysis
Statistical analysis of this in vitro study was performed using ANOVA followed by Tukey's post-hoc test for metabolic activity data and Kruskal-Wallis test followed by Dunn's post-hoc test for checkerboard data (p≤0.05).
6.Results
The graph in Figure 3 needs a more descriptive title.
Author´s Answer: Figure 3 titles were modified as suggested to: “ Mean counts of each bacterial species (x105)”
7.Discussion
In lines 225 – 236, the virucidal effect of methylene blue is described, and no apparent relation is established with the results observed in bacterial biofilms. On the other hand, their effect on bacteria is only mentioned without comparison with the presented results and a description of the mode of action. Despite planktonic cells and biofilm former ones presenting different susceptibilities to antimicrobial agents, studies performed with planktonic cultures provide insight into their mode of action. Thus, giving more detail of the mentioned studies in lines 234 – 236 can help explain the observed reduction in the metabolic activity of biofilm-formed cells. This observation also applies to the effect of LED treatment and its combination with methylene blue; thus, it is recommended to include and discuss the outcomes of studies that implemented similar approaches in oral bacteria.
Author´s Answer: Thanks for the suggestion that improved our manuscript. Information regarding possible mechanism of action were included as follow:
However, studies with planktonic bacteria may be helpful to reveal mechanisms of action. In this sense, the literature reports that as the photosensitizer pH increases, greater P. gingivalis inhibition may be found [44] . Furthering, MB may bind to P. gingivalis cell surface and then, after absorption of the light (600- to 700-nm wavelength) with peak, the photosensitizer produces reactive oxygen species, and injury several bacterial proteins, lipids, and carbohydrates leading to bacterial cell death [47]. Thus, future studies should confirm whether theses mechanisms still occur within subgingival biofilms.
- Lines 266 – 274 and 283 – 289 contain the same information described in lines 257-265 and 276-289, respectively.
Author´s Answer: Thank you very much. The repetition was removed from the manuscript.
Reviewer 2 Report
Comments and Suggestions for Authors
In this paper, the authors study in vitro in biofilm the effect of one photosensitizer, the commercial methylene blue, red LED, and the combination. Proposing this photosensitizer + LED as a photodynamic therapy system against polymicrobial biofilm.
Although the authors have carried out some in vitro experiments, I recommend revising the work specifically the manuscript because in my opinion the results should be explained better.
- In the introduction section, the photodynamic therapy should be explained more clearly (lines 66 to 72).
- In the materials and methods section, in point 2.2 Biofilim treatments, more details about the experiments are required. Firstly, what is the MB concentration (Molarity)? Regarding the treatments, all the groups are carried out under the same conditions: MB concentration, time of exposition, and distance from culture. Moreover, in line 124 the unit of λ=660 does not appear, and in the power density mW/cm2, the 2 has to be in 2, superscript. The same thing happens later in the line 258.
- In the results section, the Figures must be better described. Appearing “a” and “b” terms, but I could not find what they meant. The error bars do not look good, due to are the same color as the bars.
-In the discussion part, I feel that all the information from lines 200 to 245 is introduction and conceptualization, instead of discussion. On line 201 the authors wrote: “and different photosensitizers” but they analyzed only one PS, methylene blue. In addition, I recommend adding the methylene blue photophysical properties, wavelength, singlet oxygen production, at least including this information in the manuscript and adding a bibliographic reference.
- On the other hand, in the discussion part, the lines from 260 to 265 are repeated exactly in lines 266-274.
-Apart from all of these, I feel that the innovation of the combination of MB + LED has to be emphasized, due to from my point of view only the LED (without MB) has the same photoeffect as LED+MB, so I can appreciate the photodynamic effect of the MB under irradiation.
-Finally, I recommend rewriting the conclusion.
Author Response
REVIEWER #2
In this paper, the authors study in vitro in biofilm the effect of one photosensitizer, the commercial methylene blue, red LED, and the combination. Proposing this photosensitizer + LED as a photodynamic therapy system against polymicrobial biofilm.
Although the authors have carried out some in vitro experiments, I recommend revising the work specifically the manuscript because in my opinion the results should be explained better.
- In the introduction section, the photodynamic therapy should be explained more clearly (lines 66 to 72).
Author´s Answer: We thank the reviewer for the suggestion that improve the clarity of our manuscript. We modified the requested paragraph as requested.
- In the materials and methods section, in point 2.2 Biofilim treatments, more details about the experiments are required. Firstly, what is the MB concentration (Molarity)? Regarding the treatments, all the groups are carried out under the same conditions: MB concentration, time of exposition, and distance from culture. Moreover, in line 124 the unit of λ=660 does not appear, and in the power density mW/cm2, the 2 has to be in 2, superscript. The same thing happens later in the line 258.
Author´s Answer: The concentration of MB is shown as a percentage (0.01 g in 100 mL), the usual presentation of this substance. Yes, all groups that contain LED in their treatment were performed in the same conditions. We included this information in the manuscript in Section 2.2 and modified the format of 2, as requested. Thank you.
- In the results section, the Figures must be better described. Appearing “a” and “b” terms, but I could not find what they meant. The error bars do not look good, due to are the same color as the bars.
Author´s Answer: The terms “a” and “b” indicate the statistical significance among groups as explained in the legend of figures. However, to make it clearer, we modified the legends as follow:” Different letters (“a” and “b”) represent statistically significant differences among groups by Kruskal-Wallis and Dunn’s posthoc tests (p ≤ 0.05).”
4.In the discussion part, I feel that all the information from lines 200 to 245 is introduction and conceptualization, instead of discussion. On line 201 the authors wrote: “and different photosensitizers” but they analyzed only one PS, methylene blue. In addition, I recommend adding the methylene blue photophysical properties, wavelength, singlet oxygen production, at least including this information in the manuscript and adding a bibliographic reference.
Author´s Answer: Part of the lines indicated by the reviewer were moved to introduction and duplicated information was removed form the manuscript. Thanks for the suggestion that improved our manuscript.
- On the other hand, in the discussion part, the lines from 260 to 265 are repeated exactly in lines 266-274.
Author´s Answer: Thanks for your note. The repetition was removed.
6.Apart from all of these, I feel that the innovation of the combination of MB + LED has to be emphasized, due to from my point of view only the LED (without MB) has the same photoeffect as LED+MB, so I can appreciate the photodynamic effect of the MB under irradiation.
Author´s Answer:
Thanks to the reviewer for the observation. More than the use of methylene blue and LEDs, what is innovative in this study is its use in our model of multispecies polymicrobial biofilms, since in the literature biofilms of a maximum of 2-3 microorganisms are reported and in our case there are 33 microorganisms that reflect the reality of the subgingival microbiota.
The possible explanation for the results of the effects of LED (without MB) vs LED+MB was clarified in a paragraph in the discussion and is based on the incubation or pre-irradiation time of the MB. This should be included in subsequent studies. This is the information added:
A factor that must be taken into account and that can increase the antimicrobial activity of the photosensitizer is the incubation/pre-irradiation time, since this could increase the joint effect of the MB+LED [58].
7.Finally, I recommend rewriting the conclusion.
Author´s Answer: The conclusion was rewritten as suggested.
Reviewer 3 Report
Comments and Suggestions for Authors
The paper is interesting and generally properly elaborated. In my opinion it could be published in "Pathogens" are making some minor revisions suggested below:
In "Material and Methods" section it should be clarified why the applied physical parameters of the photodynamic therapy have been selected (wavelength and of light as well as time of exposure).
In "Results" section in captions for figures 1 and 2 the meaning of letters a and b presenting the statistical significance should be described as clearly as in caption for figure 3.
In description of physical parameters of photodynamic therapy a statement (λ = 660 nm, power density ≈ 330 mW/cm2, 2 mm distance from culture) should be used instead of the statement (λ = 660, power density ≈ 330mW/cm2, 2mm distance from culture) - both in Abstract and in section 2.2 Biofilm treatments.
Comments on the Quality of English Language
Moderate editing of English language is necessary
Author Response
REVIEWER #3
The paper is interesting and generally properly elaborated. In my opinion it could be published in "Pathogens" are making some minor revisions suggested below:
In "Material and Methods" section it should be clarified why the applied physical parameters of the photodynamic therapy have been selected (wavelength and of light as well as time of exposure).
Author´s Answer: Modified as suggested.
several studies [32–34] have chosen and justified the use of LED lights with wavelengths between 450-660nm, which comprise the spectrum used in the present study. The reasons that support this are the physical characteristics of the LED, including its narrow emission spectrum that benefits the maximum absorption of the photosensitizers. Furthermore, the use of a LED source guarantees a larger irradiation field due to the lack of collimation and perfect coherence of the LEDs
In "Results" section in captions for figures 1 and 2 the meaning of letters a and b presenting the statistical significance should be described as clearly as in caption for figure 3.
Author´s Answer: Modified as suggested.
In description of physical parameters of photodynamic therapy a statement (λ = 660 nm, power density ≈ 330 mW/cm2, 2 mm distance from culture) should be used instead of the statement (λ = 660, power density ≈ 330mW/cm2, 2mm distance from culture) - both in Abstract and in section 2.2 Biofilm treatments.
Author´s Answer: Modified as suggested.
Round 2
Reviewer 2 Report
Comments and Suggestions for Authors
The authors followed all my suggestions so I consider accepting the manuscript.
Author Response
REVIEWER #2
The authors followed all my suggestions so I consider accepting the manuscript.
Author´s Answer: Thank you very much. Your suggestions improved our manuscript.